# Low Antibody Dosing in Cancer Therapy: Targeted Cytotoxicity Combined with Anti-Tumour Immunostimulation

**DOI:** 10.3390/ijms26167724

**Published:** 2025-08-10

**Authors:** Victor I. Seledtsov, Galina V. Seledtsova, Adas Darinskas, Alexei von Delwig

**Affiliations:** 1Petrovsky National Research Centre of Surgery, 119991 Moscow, Russia; delvig59@mail.ru; 2Institute for Fundamental and Clinical Immunology, 630099 Novosibirsk, Russia; galina-seledtsova@yandex.ru; 3National Cancer Institute, 08406 Vilnius, Lithuania; darinskas.adas@gmail.com

**Keywords:** therapeutic monoclonal antibody, tumor-associated antigen, growth factor receptor, PD-L1, targeted cytotoxicity, immunostimulation

## Abstract

Overexpression of growth factor receptors and immunosuppressive molecules is a hallmark of many tumour cells, distinguishing them from normal tissue. This co-expression enables tumours both to exploit proliferative signalling and to evade immune surveillance. Here, we propose a strategy that employs a combination of monoclonal antibodies (mAbs) targeting two distinct antigens (Ags) at sub-cytotoxic doses. This approach aims to achieve a threshold cytotoxic density of immune complexes selectively on malignant cells expressing both target Ags, while sparing normal cells that express only one. Typically, the first target Ag may be a growth factor receptor, such as epidermal growth factor receptor (EGFR and HER1), epidermal growth factor receptor 2 (HER2), or vascular endothelial growth factor receptor 2 (VEGFR2), and the second, an immunoinhibitory molecule, such as programmed death-ligand 1 (PD-L1). Selective mAb-mediated tumour destruction is expected to enhance neoantigen (NeoAg) presentation to the immune system, while the blockade of PD-1/PD-L1 interactions should further stimulate anti-tumour immune responses. Notably, this strategy can be implemented using clinically approved therapeutic mAbs, potentially enabling rapid translation into clinical practice without extensive regulatory hurdles.

## 1. Introduction

Despite significant advances in systemic chemotherapy, the lack of selectivity of most anti-cancer agents remains a critical limitation. Conventional cytotoxic drugs not only target malignant cells but also affect rapidly dividing healthy cells—such as those in the bone marrow, gastrointestinal tract, and hair follicles—resulting in substantial toxicity and adverse effects [1]. Efforts to develop chemotherapeutic agents with tumour-selective activity have largely been unsuccessful, owing to the biochemical and molecular similarities between cancerous and normal cells, particularly within essential metabolic and proliferative pathways [2].

Unlike their intracellular homogeneity, tumour cells often differ from their normal counterparts in surface antigenic composition. These differences can trigger immune recognition and response, positioning the immune system as a powerful tool for tumour control. However, cancer progression frequently involves immune escape mechanisms, including upregulation of immune checkpoint molecules (e.g., PD-L1), recruitment of regulatory T-cells, secretion of immunosuppressive cytokines, and remodelling of the tumour microenvironment [3,4]. Additionally, immunosenescence—the age-related decline in immune function—impairs the immune system’s ability to combat tumours and reduces the efficacy of immunotherapies such as cancer vaccines and cytokine treatments, particularly in older patients [5].

Consequently, passive immunotherapeutic approaches—especially those utilising monoclonal antibodies (mAbs)—have become a cornerstone of modern oncology, complementing surgery, radiotherapy, and chemotherapy. Therapeutic mAbs exhibit a range of clinically relevant mechanisms of action, directly targeting tumour cells while simultaneously promoting anti-tumour immune responses [6]. Their multifaceted properties have spurred the development of novel cancer treatment strategies with significant potential to transform patient care. Although combinatorial mAb-based regimens hold great promise, their increased systemic toxicity often limits their clinical application [1,7].

Tumour cells often simultaneously overexpress growth-promoting and immunosuppressive surface molecules, a pattern rarely observed in normal tissues. This co-expression creates a unique therapeutic opportunity to selectively target cancer cells using dual-specificity approaches. By leveraging combinations of mAbs that bind differentially expressed tumour-associated antigens (TAAs), it may be possible to enhance selectivity and reduce off-target effects. Importantly, this approach aligns with current trends in precision oncology, where treatment is increasingly tailored to tumour-specific molecular profiles. The feasibility of stratifying tumours based on expression patterns of both growth factor receptors (e.g., EGFR, HER2, and VEGFR2) and immune checkpoint molecules (e.g., PD-L1) suggests that this strategy could be applied across multiple cancer types. In this article, we present a dual-targeting approach aimed to expand the therapeutic window of mAb-based immunotherapy by combining selective cytotoxicity with immune activation, while minimising the risk of severe, life-threatening adverse events.

## 2. Tumour-Associated Antigens and Challenges of Targeted Immunotherapy

TAAs are broadly categorised into two main groups: (i) tumour-specific neoantigens (NeoAgs)—unique peptides encoded by mutated or viral genes, expressed exclusively by malignant cells; and (ii) shared antigens (Ags)—those that may also be present on normal tissues, often in a restricted, developmental stage-dependent, or low-level constitutive manner [8,9].

The tumour-specific expression of NeoAgs makes them attractive targets for highly selective immunotherapy [10,11]. However, their practical application is hampered by several challenges. First, different tumours—even within the same tissue type—may generate distinct NeoAg repertoires. Second, intra-tumoural heterogeneity means that NeoAg expression may be limited to only a subset of malignant clones. As a result, elimination of NeoAg-expressing cells often fails to halt tumour progression, allowing immune escape through the expansion of NeoAg-negative clones [12,13]. Furthermore, due to the random nature of mutation-derived Ags, most NeoAgs are private (patient-specific), necessitating personalised vaccine or cell therapy strategies that are costly and time-consuming to develop [14]. Although computational methods and high-throughput sequencing technologies have improved the prediction and identification of NeoAgs, validation of their immunogenicity remains labourious and uncertain.

In contrast to NeoAgs, shared TAAs—representing the majority of known TAAs—are typically normal cellular proteins that are either overexpressed in tumours or aberrantly expressed in contexts where they are normally silent [15]. These include various growth factor receptors, many of which play pivotal roles in maintaining tumour growth and survival by interacting with physiological growth signals present in the tumour microenvironment. Notably, their overexpression is often stable across the tumour cell population, making them relatively consistent targets for mAb-based therapies [16].

A distinct subset of shared TAAs consists of immunoregulatory membrane proteins, including checkpoint ligands such as programmed death-ligand 1 (PD-L1), cytotoxic T-lymphocyte-associated protein 4 (CTLA-4), and Galectin-9, which enable tumour cells to suppress T-cell-mediated cytotoxicity and evade immune destruction [17,18]. This dual co-overexpression of proliferative and immunosuppressive molecules underlies the aggressive phenotype of many cancers and contributes to therapeutic resistance [19].

## 3. Therapeutic Monoclonal Antibodies in Cancer Treatment: Mechanisms and Classification

Therapeutic mAbs have become a cornerstone of contemporary treatment strategies for a wide range of diseases, particularly cancer. Modern bioengineering technologies allow for the production of low-immunogenic cytotoxic IgG Abs with virtually any specificity. As of 15 May 2024, 76 anti-cancer mAbs have been approved by World Health Organization (WHO)-Listed Authorities (WLA) across multiple countries [20]. These agents are distinguished by their high target specificity and multifaceted mechanisms of action, enabling potent anti-tumour responses while reducing systemic toxicity and adverse effects compared to conventional chemotherapies [21]. Their modular structure also allows for extensive engineering to enhance pharmacokinetics, tissue penetration, and immune effector function, contributing to the growing success of Ab-based therapies in both solid and haematological malignancies.

Immunoglobulins are divided into five classes: IgA, IgD, IgE, IgG, and IgM. Among them, IgG is the predominant class used in mAb therapy, largely due to its compatibility with Fc gamma receptors (FcγRs) on natural killer (NK) cells, macrophages, dendritic cells, neutrophils, and eosinophils. The IgG class is further divided into four subclasses—IgG1, IgG2, IgG3, and IgG4—each varying in its ability to mediate Ab-dependent cellular cytotoxicity (ADCC), Ab-dependent cellular phagocytosis (ADCP), complement-dependent cytotoxicity (CDC), and complement-dependent cellular cytotoxicity (CDCC) [22,23]. IgG1 and IgG3 exhibit robust effector functions, particularly in activating ADCC and CDC pathways. However, IgG3 is less favoured therapeutically due to its long hinge region, which is more susceptible to proteolysis and immunogenic polymorphisms [23,24]. Consequently, IgG1 is the most widely used subclass in oncology, owing to its strong binding to activating FcγRs and its ability to mediate ADCC, ADCP, and CDC, whereas the IgG2 and IgG4 subclasses, which display weaker Fc-mediated effector functions, and are generally reserved for neutralising functions or when reduced immunogenicity is desirable—for example, in chronic inflammatory diseases or when targeting non-lethal tumour-associated pathways [25].

ADCC and ADCP occur when effector cells—such as NK cells, granulocytes, and macrophages—bind to tumour cells coated with Abs via their Fc receptors, initiating downstream cytodestructive pathways. In ADCC, NK cells release perforin and granzymes, leading to target cell lysis. In ADCP, phagocytes engulf and digest the Ab-coated tumour cells. The efficiency of ADCC is influenced by several factors, including the level and density of target Ag expression, the mAb isotype, and the administered mAb dose, all of which correlate with clinical response [26].

CDCC, by contrast, requires prior activation of the complement system. This results in the deposition of complement components such as C1q, C3b, iC3b, or C4b on the target cell surface, thereby facilitating Fc-mediated recruitment and destruction by effector cells. CDC is initiated when IgM or IgG Abs bind to cell-surface Ags, triggering the classical complement pathway—a cascade involving more than 30 serum proteins that culminates in the formation of the membrane attack complex and subsequent lysis of tumour cells [27].

mAbs form immune complexes with target Ags from damaged cells, which are efficiently internalised by antigen-presenting cells (APCs) via FcγRss. This leads to enhanced MHC I and MHC II cross-presentation, improving CD8^+^ and CD4^+^ T-cell responses [9]. Sustained antigen presentation prevents T-cell exhaustion or depletion by maintaining activation signals [26].

All therapeutic anti-tumour mAbs can be broadly classified into the following two groups: (i) direct tumour-destructive mAbs, which inhibit tumour growth by targeting surface TAAs, and (ii) immunostimulatory mAbs, which target immunoinhibitory molecules (e.g., PD-1 and CTLA-4), thereby enhancing anti-tumour immune responses. Table 1 presents widely used cytotoxic mAbs in the treatment of solid tumours.

For proper understanding of the therapeutic approach described below, it is important to highlight that all mAbs listed in Table 1 possess dual functionality: they are capable of both direct tumour cell killing and promoting anti-tumour immune responses through immunogenic antigen presentation and T-cell activation [23].

## 4. Designing Low-Dose, Multi-Target mAb Therapies for Treatment of Solid Tumours

A significant challenge in the therapeutic application of cytotoxic mAbs is the expression of TAAs, not only on tumour cells, but also on normal cells. However, due to their genetic and epigenetic instability, tumours often exhibit abnormal membrane co-expression of antigenic molecules, which markedly increases their sensitivity to growth stimuli while simultaneously reducing their recognisability by immune cells [28]. This aberrant co-expression enhances tumour aggressiveness and renders them less susceptible to immune surveillance. Conversely, the unusual co-expression of Ags on tumour cells makes them more vulnerable to the cytotoxic effects of combinations of therapeutic Ag-specific mAbs [29].

From a conventional standpoint, the cytotoxic effect mediated by Abs is characterised by a threshold phenomenon, which suggests that immune complexes bound to cell membranes must reach a specific density to initiate cytolysis [29]. This concept has been supported by experimental evidence involving both polyclonal and monoclonal Abs targeting a range of membrane-associated Ags [30,31]. The biological significance of this threshold has been postulated as a protective mechanism to avoid excessive cytotoxicity from minimal antigenic stimuli that are harmless to the organism [32]. This principle underpins the rationale for polyantigenic anti-cancer vaccines, which demonstrate more robust anti-tumour clinical effects than their monoantigenic counterparts [9].

Evidently, the probability of reaching the cytolytic threshold density of membrane-associated immune complexes on tumour cells increases when multiple therapeutic mAgs targeting distinct TAAs are used in combination. A novel immunotherapeutic concept—low-dose Ab immunotherapy—has been proposed, wherein more than one membrane Ag-specific Ab is administered at suboptimal dosages to selectively target tumour cells while sparing normal tissue [29]. This approach enables the formation of Ag–Ab complexes at cytolytic density thresholds only on tumour cells that co-express multiple target Ags. In contrast, normal cells, lacking such co-expression, do not accumulate sufficient immune complex density to trigger cytotoxicity. As such, two or more mAbs may be used at suboptimal (sub-cytotoxic) doses, reducing the risks of complications associated with a single mAb at an optimal (cytotoxic) dose.

Figure 1 illustrates potential combinations of target Ags that can be exploited simultaneously for mAb-mediated tumour cell killing and stimulation of anti-tumour immune responses. It demonstrates the concept that, while a single mAb preparation may be insufficient to induce lysis, the concurrent use of two mAbs—each recognising a distinct membrane-associated Ag—can generate immune complexes dense enough to trigger cytolytic mechanisms. Thus, selective and effective tumour destruction may be more reliably achieved through the combinatorial use of low-dose TAA-specific mAbs, rather than relying on a single high-dose Ab against a single TAA.

Table 2 presents potential target combinations of Ag1 and Ag2 for the combined use of corresponding mAbs at low (sub-cytotoxic) doses.

The ERBB receptor family, also known as the epidermal growth factor (EGF) receptor or type I receptor family, comprises ERBB1/HER1, ERBB2/HER2, ERBB3/HER3, and ERBB4/HER4. Heterodimerisation is a key activation mechanism for all ERBB receptors in response to ligand binding. Activated ERBB receptors interact with numerous signalling proteins and initiate pathways governing cell proliferation, angiogenesis, apoptosis, and metastatic spread. However, current clinical inhibitors of these receptors (e.g., erlotinib, lapatinib, gefitinib, and selumetinib) lack specificity and are associated with undesirable adverse effects [33].

Cytotoxic mAbs targeting ERBB1/HER1 and ERBB2/HER2 are widely used in cancer therapy. Their frequent overexpression in solid tumours makes them logical candidates for Ag 1 (growth factor receptor) in this combinatorial approach involving sub-cytotoxic mAb doses. Another candidate is vascular endothelial growth factor receptor-2 (VEGFR-2), a key mediator of tumour angiogenesis, also expressed across various malignancies. Notably, HER2 overexpression is linked with poor prognosis in breast and gastric cancers, and anti-HER2 mAbs (e.g., trastuzumab) are well established in the treatment of HER2-positive breast cancer [34].

However, as with HER1, HER2 expression is not exclusive to malignant tissues. Its low-level expression in normal organs raises concerns about on-target/off-tumour toxicity, limiting broader application of HER-targeted therapies [35].

For Ag2 (immunoinhibitory molecule), PD-L1 is an optimal candidate. PD-L1 suppresses T-cell activation and facilitates tumour progression. Its overexpression in cancers, such as gastric, hepatocellular, renal, oesophageal, pancreatic, ovarian, and bladder cancers, is associated with poor prognosis [36]. In practice, anti-PD-L1 mAbs are administered as immunostimulatory agents at sub-cytotoxic doses. Higher doses have been associated with severe, life-threatening immune-related complications. Moreover, depletion of activated PD-L1-positive T-lymphocytes by these mAbs may result in long-term immune dysfunction [37].

Preclinical and clinical studies have shown that HER2-positive breast tumours can also express PD-L1, contributing to immune evasion and resistance to HER2-targeted therapy. In experimental models, a humanised bispecific IgG1 subclass Ab targeting both HER2 and PD-L1 (HER2/PD-L1 BsAb) demonstrated strong therapeutic activity directed against HER2- and PD-1-positive tumour cells [38]. However, we note that in this particular work, the role of the anti-PD-L1 component is primarily considered immunostimulatory (non-cytotoxic), aiming to promote T-cell activation. In our view, one of the advantages of using separate (non-conjugated) mAbs lies in the ability to optimise their concentrations independently based on the expression levels of each target marker within the tumour. Such separate optimisation is not possible when using bispecific Abs.

Combination treatments targeting both HER2 and immune checkpoints are under investigation in clinical trials to overcome immune evasion and enhance anti-tumour immune responses in HER2-positive breast cancer [39,40]. However, these studies have used anti-HER2 mAbs at optimal (cytotoxic) doses and did not explore the direct anti-tumour cytotoxic effects of anti-PD-L1 mAbs. Similar limitations are seen in clinical studies involving anti-HER1 and anti-PD-L1 combinations [41,42]. To date, the therapeutic potential of reduced (suboptimal) doses of anti-HER and anti-PD-L1 mAbs remains unexplored.

Evidence suggests that both HER1 and HER2 signalling negatively regulate anti-tumour immunity. HER activation can downregulate MHC class I/II expression on tumour cells, impairing T-cell recognition [40,43]. ERBB1 ligands (e.g., EGF) can induce IL-6 and IL-8, promoting chronic inflammation and tumour progression [44,45]. HER2 signalling attracts immunosuppressive cells such as regulatory T-cells (Tregs), myeloid-derived suppressor cells (MDSCs), and tumour-associated macrophages (TAMs), fostering an immunosuppressive TME [46]. It also upregulates chemokines like CCL2 and CXCR4, promoting a pro-inflammatory and immunosuppressive environment conducive to tumour growth and metastasis. HER2 activation further induces PD-L1 expression on tumour cells, leading to PD-1-dependent T-cell exhaustion and immune evasion [40].

VEGF/VEGFR2 signalling also plays a role in immune regulation. Inhibition of this pathway can normalise tumour vasculature, increase intra-tumour infiltration of lymphocytes, and decrease the number and function of inhibitory immune cells (e.g., MDSCs, Tregs, and M2 macrophages) [46,47].

Accordingly, HER1, HER2, and VEGFR2 can serve dual roles as growth factor receptors (Ag 1) and immunoinhibitory molecules (Ag 2). The combined use of anti-HER1 and anti-HER2, or anti-HER and anti-VEGFR2, mAbs has been investigated across various cancers. While promising, these studies have demonstrated high toxicity, underscoring the need for treatment optimisation [48,49].

The proposed concept can be extended to include more than two mAb therapeutics, with the theoretical advantage that increasing the number of target Ags allows for proportional reductions in individual mAb doses. This may reduce on-target/off-tumour toxicity commonly observed with high-dose monotherapies. Critically, the doses of each mAb within such combination regimens must be optimised to ensure selective tumour killing while sparing normal tissue. Several platforms now exist to define threshold and sub-threshold Ab concentrations for both experimental and translational applications [50,51,52,53]. The higher the Ag expression on tumour cells, the lower the required dose of the corresponding mAb, and vice versa. Thus, combination mAb therapy should be based on both qualitative and quantitative assessment of tumour antigenic composition. We hypothesise that anti-tumour mAb combinations will exhibit a defined therapeutic window, akin to traditional pharmacological agents, with each mAb dose calibrated to maximise anti-tumour efficacy while minimising toxicity to healthy tissues.

Additionally, as with other drug regimens, mAb combination dosages would need to be tailored to the patient’s disease severity and condition. Continuous monitoring of patient responsiveness to immunomodulatory therapy is essential. Immune cell dysfunctions, complement depletion, and the presence of IgG-containing immune complexes in patients’ serum can reduce mAb-mediated cytotoxicity [54]. This clearly necessitates that passive mAb-based immunotherapy regimens be highly personalised, incorporating a comprehensive set of supportive measures—such as transfusion of donor leukocytes and/or fresh, complement-sufficient plasma preparations—to maximise clinical effectiveness.

Our concept suggests that therapeutic mAbs conjugated with artificial cytotoxic agents could also be applied for selective tumour destruction. The key requirement for such agents should be the existence of a quantifiable threshold for triggering their cytotoxic effect [55].

The concept of using low-dose mAbs in anti-cancer therapy, as outlined in this article, does have a significant limitation. Not all tumour cells—even within the same tumour—co-express the target growth receptors and immunosuppressive molecules. As a result, these cells may evade the cytotoxic effects of low-dose co-administered mAbs. Nevertheless, the cytotoxic activity of mAbs against the most aggressive tumour cells that do co-express the target molecules, combined with their immunostimulatory properties, should be sufficient to halt or at least inhibit overall tumour growth.

## 5. Conclusions

The immunotherapeutic strategy proposed here may be regarded as an immunological scalpel—a precision-guided approach designed to selectively eliminate cancer cells while preserving the integrity of normal tissues and stimulating anti-tumour immune responses. By directing cytotoxic activity specifically towards TAAs and avoiding collateral damage to healthy cells, this strategy marks a significant advancement in the evolution of mAb-based cancer therapies. Leveraging therapeutic mAbs already approved for clinical use, it presents a feasible and potentially regulatory-friendly pathway towards more refined anti-cancer immunotherapeutic interventions. Furthermore, the use of these mAbs at low doses not only minimises the risk of off-target effects but also improves the cost-effectiveness and scalability of treatment.

## Figures and Tables

**Figure 1 ijms-26-07724-f001:**
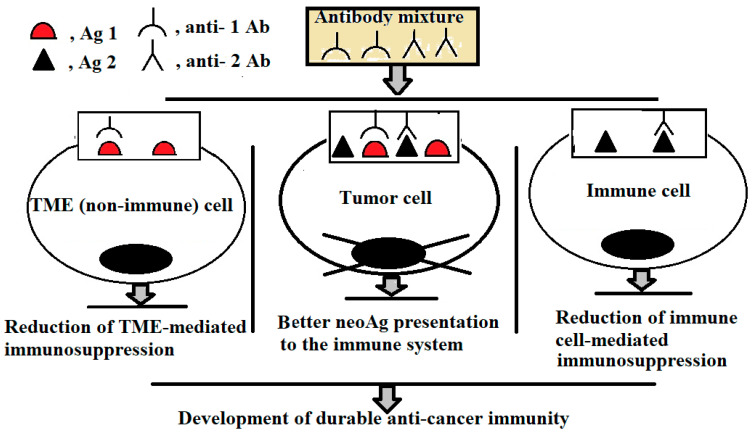
Schematic representation of the combined action of sub-cytotoxic mAb doses on cells. Anti-Ag1 and anti-Ag2 Abs bind to Ag1 (growth factor receptor) and Ag2 (immunoinhibitory molecule), respectively, reaching the cytotoxic threshold of immune complex formation on tumour cells. The Abs are unable to exert cytotoxic effect on non-tumour cells expressing only Ag1 or Ag2. Their interactions with non-tumour cells (both immune and non-immune) attenuates immunosuppressive activity of tumour microenvironment (TME).

**Table 1 ijms-26-07724-t001:** FDA-approved monoclonal cytotoxic antibodies for solid cancer.

Name, IgG Subclass	Antigen	Indications (Tumour Types)
Cetuximab, chimeric IgG1	Epidermal growth factor receptor (HER1)	Colorectal cancer (2004); head and neck squamous cell carcinoma (2006)
Necitumumab, human IgG1	HER1	Non-small-cell lung cancer (2015)
Trastuzumab, humanized IgG1	Epidermal growth factor receptor 2 (HER2)	Breast cancer (1998)
Pertuzumab, humanized IgG1	HER2	Breast cancer (2012)
Ramucirumab, human IgG1	Vascular endothelial growth factor receptor 2 (VEGFR2)	Gastric cancer (2014)
Atezolizumab, humanized IgG1	Programmed death-ligand 1 (PD-L1)	Bladder cancer, non-small-cell lung cancer (2016); triple-negative breast cancer (2019)
Avelumab, human IgG1	PD-L1	Urothelial carcinoma, Merkel cell carcinoma (2017)
Durvalumab, human IgG1	PD-L1	Bladder cancer (2017)

**Table 2 ijms-26-07724-t002:** Target Ag1 (growth factor receptor) and Ag2 (immunoinhibitory molecule) for combined application of the respective mAbs at low (sub-cytotoxic) doses.

Ag1	Ag2	Target Tumours
Epidermal growth factor receptor (ERBB1, HER1)	PD-L1	Bladder cancer; breast cancer; colorectal cancer; gastroesophageal cancer; glioblastoma; head and neck squamous cell carcinoma (HNSCC); non-small-cell lung cancer (NSCLC); ovarian cancer
Human epidermal growth factor receptor 2 (ERBB2, HER2)	PD-L1	Bladder cancer; breast cancer; colorectal cancer; endometrial cancer; gastroesophageal cancer, NSCLC; ovarian cancer; pancreatic cancer
Vascular endothelial growth factor receptor-2 (VEGFR-2)	PD-L1	Breast cancer; colorectal cancer; gastroesophageal cancer; glioblastoma; hepatocellular carcinoma; NSCLC; ovarian cancer; renal cell carcinoma; soft tissue sarcoma; thyroid cancer
ERBB1 (HER1)	VEGFR-2	Breast cancer; colorectal cancer; gastroesophageal cancer; glioblastoma; NSCLC; ovarian cancer
ERBB2 (HER2)	VEGFR-2	Breast cancer; colorectal cancer; gastroesophageal cancer; NSCLC
ERBB1 (HER1)	ERBB2 (HER2)	Bladder cancer; breast cancer; colorectal cancer; endometrial cancer; gastroesophageal cancer; NSCLC; ovarian cancer

Note: Target tumours were identified based on the tumour-associated co-expression of Ag1 and Ag2 using the DeepSeek programme.

## Data Availability

Data sharing requires no permission.

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
