# Peer review of "Low Antibody Dosing in Cancer Therapy: Targeted Cytotoxicity Combined with Anti-Tumour Immunostimulation"

_ijms, 2025, doi:10.3390/ijms26167724_

Round 1

Reviewer 1 Report

Comments and Suggestions for Authors

This manuscript summarizes the strategy of combination of monoclonal antibodies targeting epidermal growth factor receptor (EGFR, HER1), epidermal growth factor receptor (HER2), or vascular endothelial growth factor receptor 2 (VEGFR2), and programmed death-ligand 1 (PD-L1).

There are still some concerns.

(1) The combination of antibody or antibody conjugation has been reported. For example, a humanized bispecific IgG1 subclass antibody targeting both HER2 and PD-L1 (HER2/PD-L1; BsAb), which displayed satisfactory purity, thermostability, and serum stability, has been constructed to direct the anti-PD-L1 responses toward HER2-expressing tumor cells, with the purpose to simultaneously target both HER2 and PD-1/PD-L1 signaling pathways. (A bispecific antibody targeting HER2 and PD-L1 inhibits tumor growth with superior efficacy, Journal of Biological Chemistry, 2021, DOI: 10.1016/j.jbc.2021.101420). It might also be worthy for the authors to highlight the advantage/weakness of antibody mixture, antibody-conjugation, neo-antibody, or nano-antibodies, in the case of immunotherapy.

(2) Besides to antibody conjugation or mixture, antibody-drug conjugates might also be helpful in tumor immunotherapy. For example, a novel HER2-targeting antibody-drug conjugates combined with immune checkpoint inhibitors can achieve remarkable effects in mice and elicit long-lasting immune protection in a hHER2+ murine breast cancer model, which provides insights into the efficacy of RC48 therapeutic activity and a rationale for potential therapeutic combination strategies with immunotherapy (Breast Cancer Res Treat, 2022, 191(1):51-61. doi: 10.1007/s10549-021-06384-4). The authors might illustrate the advantage of this manuscript, in comparison with other reports.

(3) The antibody’s structure/sequence design is very important for immunotherapy. The authors might be worthy to provide this information.

(4) If there is any clinical trials, please provide the detailed therapeutic outcome, in comparison with single therapy only.

(5) For table 2, for the target tumors, it might be better to list the tumor type in order (either by the first letter or therapeutic outcome), which might be easier for readers to follow.

Reviewer 2 Report

Comments and Suggestions for Authors

Firstly, please uniform every table with the three-line format;

Although it was mentioned that the use of approved mAbs can accelerate clinical translation, the specific implementation challenges (such as dose optimization, patient stratification criteria, and toxicity risks of combination therapy) were not discussed;

The mechanism description of immune stimulation effect is relatively vague, without detailed explanation of how low-dose mAbs synergistically enhance new antigen presentation or overcome T cell exhaustion;

The issue of potential drug resistance is crucial, and the authors only briefly mentions that tumor heterogeneity may lead to some cells escaping, but does not discuss the specific mechanisms of drug resistance;

The logical coherence of some paragraphs is not smooth enough (such as suddenly shifting from "TAAs classification" to "mAb mechanism");

There is only one figure; the figures can help the authors present key summary and details more clearly. It is suggested to display HER2/PD-L1 co expression in different cancers using heat maps, or illustrate the synergistic mechanisms of combined mAb;

It is suggested to expand the discussion section, such as discussing individualized treatment needs (such as immune cell function monitoring) or cost-benefit analysis, as well as the potential for combination with other immunotherapies (such as CAR-T);

How to screen suitable patient populations is also an issue that needs attention.

Author Response

Please enter "Please see the attachment.

Reviewer 3 Report

Comments and Suggestions for Authors

A perspective manuscript entitled “Low Antibody Dosing in Cancer Therapy: Targeted Cytotoxicity Combined with Anti-Tumour Immunostimulation” by V.I. Seledtsov, G.V. Seledtsova, A. Darinskas and A. von Delwig reviews immunotherapeutic approaches using monoclonal antibodies and proposes a novel strategy combining sub-cytotoxic doses of monoclonal antibodies targeting two distinct antigens. This study aligns well with the scope of the International Journal of Molecular Sciences and is suitable for publication after minor revisions.

I recommend the authors expand their review with recent works.

Introduction. “Consequently, passive immunotherapeutic approaches—especially those utilizing monoclonal antibodies (mAbs)—have become a cornerstone of modern oncology, complementing surgery, radiotherapy, and chemotherapy” please add examples.

Are there other similar studies or simulations that confirm the validity of the proposed concept?

Author Response

(The authors gave the same response as above.)

Round 2

Reviewer 2 Report

Comments and Suggestions for Authors

The authors have made corresponding revisions based on the review comments, and it is recommended that the paper is accepted in its current status.    
